# Could Biparametric MRI Replace Multiparametric MRI in the Management of Prostate Cancer?

**DOI:** 10.3390/life13020465

**Published:** 2023-02-07

**Authors:** Roxana Iacob, Emil-Robert Stoicescu, Simona Cerbu, Diana-Luminiţa Manolescu, Răzvan Bardan, Alin Cumpănaş

**Affiliations:** 1Doctoral School, “Victor Babes” University of Medicine and Pharmacy, 300041 Timisoara, Romania; 2Discipline of Radiology and Medical Imaging, “Victor Babes” University of Medicine and Pharmacy Timisoara, Eftimie Murgu Square No. 2, 300041 Timisoara, Romania; 3Research Center for Pharmaco-Toxicological Evaluations, “Victor Babes” University of Medicine and Pharmacy Timisoara, Eftimie Murgu Square No. 2, 300041 Timisoara, Romania; 4Center for Research and Innovation in Precision Medicine of Respiratory Diseases (CRIPMRD), “Victor Babeș” University of Medicine and Pharmacy, 300041 Timișoara, Romania; 5Discipline of Urology, “Victor Babes” University of Medicine and Pharmacy Timisoara, Eftimie Murgu Square No. 2, 300041 Timisoara, Romania

**Keywords:** prostate cancer, multiparametric MRI, biparametric MRI, imaging techniques, diagnosis, mpMRI, bpMRI

## Abstract

**Simple Summary:**

Prostate cancer (PCa) is one of the most common types of tumors among men, being potentially curable when diagnosed in early phases. With all this, this type of cancer does not have any specific symptoms until it advances. Therefore, it is vital to diagnose it as soon as possible. This study aims to determine the role of biparametric MRI (bpMRI) in PCa management, compared to today’s most used imaging method—multiparametric MRI (mpMRI). There were 31 relevant articles in the field studied to find the advantages and disadvantages of bpMRI. The results show that bpMRI can mostly be used for PCa diagnosis, but for other steps of tumor management, it is not as suitable as mpMRI, having a series of disadvantages. In conclusion, bpMRI has similar accuracy as mpMRI in the diagnosis of PCa. These findings will be useful in the field of PCa management, because, if bpMRI replaces mpMRI at least in diagnosis, the costs for this imaging method will be lower, and patients will be less likely to develop allergic reactions to contrast agents, as bpMRI does not use any.

**Abstract:**

Prostate cancer (PCa) is a worldwide epidemiological problem, since it is one of the most prevalent types of neoplasia among men, and the third-leading cause of cancer-related deaths, after lung and colorectal tumors. Unfortunately, the early stages of PCa have a wide range of unspecific symptoms. For these reasons, early diagnosis and accurate evaluation of suspicious lesions are crucial. Multiparametric MRI (mpMRI) is currently the imaging modality of choice for diagnostic screening and local staging of PCa, but also has a leading role in guiding biopsies and in treatment biparametric MRI (bpMRI) could partially replace mpMRI due to its lack of adverse reactions caused by contrast agents, relatively lower costs, and shorter acquisition time. Further, 31 relevant articles regarding the advantages and disadvantages of the aforementioned imaging techniques were scanned. As a result, while bpMRI has comparable accuracy in detecting PCa, its roles in the other steps of PCa management are limited.

## 1. Introduction

Prostate cancer (PCa) is one of the most common types of tumors among men worldwide, accounting for the third leading cause of deaths, after lung and colorectal tumors [1]. Even though men are less likely to die from PCa, the majority of deaths result from metastases in the spinal cord, rectum, brain, bones, or bladder [1,2]. Taller height, lipid levels, obesity, and smoking are the most common risk factors, with the strongest evidence [3,4,5].

The prostate is located within the subperitoneal, between the pubic symphysis and the rectum, and has a cone shape [6]. The presence of testosterone maintains its size and function, whereas in aging men, the gland becomes hyperproliferative and develops a predisposition to carcinoma [7,8]. It is well known for its support role in the male reproductive system. The main function of the gland is to secrete an alkaline solution, which protects the sperm from the acidic environment of the vagina [9]. The gland is divided into four zones—the peripheral zone (PZ)—the largest one, the central zone (CZ)—which accounts for 25% of the glandular tissue, the transition zone (TZ), and the anterior fibromuscular stroma [6,10]. The majority of PCa (up to 70%) develops from PZ, while 20% develops from TZ. Only 1% of this neoplasia develops in the CZ [11].

Unfortunately, small, localized PCa does not cause any symptoms, until it has progressed too far for curative treatment [12]. The non-specific symptoms such as incontinence, nocturia, and increased urinary frequency may be also caused by other non-malignant prostate pathologies, such as benign prostate hyperplasia (BHP) [13,14]. As a result, screening is extremely important [12]. Elevated blood levels of prostate-specific antigen (PSA) were used for PCa screening, but it is not a very specific test, with high levels being found also in prostatitis, BPH, physical activity, and other conditions [12,15]. 

Until quite recently, ultrasound was the most commonly used imaging method for evaluating the prostate, but it is now mostly used to guide biopsies [16]. Ultrasound examination and biopsy guiding can be performed either transrectally (TRUS), or trans-perineally, both with advantages and disadvantages [17]. Asides from ultrasound imaging, positron emission tomography and Computed Tomography (PET-CT) can be used to assess PCa, and target metabolic and cellular activities during tumor growth [18]. Multiparametric magnetic resonance imaging (mpMRI) has become the most commonly used imaging method for evaluating the prostate in recent years [16]. Although MRI was the best method for staging and detecting extracapsular extension of PCa, new innovative parameters can now identify and localize the lesion with accuracy [16,19]. Studies show that mpMRI can reduce the overdiagnosis of insignificant lesions [20]. This method, as the name implies, includes multiple sequences, such as T1-weighted images, T2-weighted images, diffusion-weighted images (DWI), and dynamic contrast-enhanced imaging (DCEI/DCE) [20,21]. The prostate can be evaluated anatomically and functionally using all of these sequences.

All of the imaging tools mentioned above are useful not only for screening and diagnosis of PCa, but also for the entire management process, which includes: screening, diagnosis, staging, treatment, and follow-up—scanning for recurrence. In terms of treatment, MRI can play a role in active surveillance (AS) [22,23]. AS is a treatment choice for low-risk, localized prostatic lesions that consists of regular hospital visits for PSA tests, MRI scans, and/or prostatic biopsies [22].

For a more standardized imaging interpretation of prostate lesions, the European Society of Urogenital Radiology (ESUR) created the prostate imaging reporting and data system (PI-RADS), which currently is in version 2 (v2) [24,25]. The PI-RADS score is a useful tool for improving the early diagnosis and treatment of PCa. The PI-RADS v2 for mpMRI can be found in the scheme below (Figure 1). There are 1–5 points that have to be assigned for each lesion, depending on how suspicious it appears:PI-RADS score 1: Very low (clinically significant cancer highly unlikely);PI-RADS score 2: Low (clinically significant cancer unlikely);PI-RADS score 3: Intermediate (clinically significant cancer equivocal);PI-RADS score 4: High (clinically significant cancer likely);PI-RADS score 5: Very high (clinically significant cancer highly likely) [26].

Biparametric MRI (bpMRI), on the other hand, uses only two types of sequences—T2W and DWI—and does not use contrast agents. As a result, the acquisition time is shorter, and the costs are lower [27]. Unlike mpMRI, bpMRI does not use contrast agents, which reduces costs, and eliminates the risk of allergic reactions and other medical conditions associated with contrast-based imaging techniques [28].

The purpose of this study is to determine whether bpMRI can be used instead of mpMRI and what its benefits and drawbacks are. As bpMRI is less expensive than mpMRI and takes less time, its limitations in the management of PCa are essential to understanding when it can be used without missing any important pathology-related details.

## 2. Materials and Methods

Using PubMed, Cochrane Library, Google Scholar, and Scopus, we selected the most important studies in the field, that compare mpMRI with bpMRI, but also evaluate bpMRI alone. After using different filters and after a manual selection of the papers, we selected 31 articles eligible for the study. The entire process of article selection is presented below.

PRISMA (Preferred Reporting Items for Systematic Reviews and Meta-Analyses) guidelines were used to select the studies for this review [29]. The current review is based on bibliographic searches in the PubMed database, Google Scholar, Cochrane Library, and Scopus. The searches were conducted manually as well as using MeSH terms (on the PubMed database). Initially, papers published in the last 5 years that were related to MRI use in the prostate cancer, specifically papers that included a comparison of bp- and mpMRI, were selected. The most relevant articles were chosen based on their title, abstract, and a quick read of the entire manuscript. Papers published in languages other than English, papers with only abstracts available, and duplicates were all excluded. First, the articles were manually searched using the keywords “prostate cancer” and each of the imaging methods individually—“multiparametric MRI”, “biparametric MRI”, and “multiparametric vs. biparametric MRI”. Following this, another PubMed search was conducted using MeSH terms, with the following keywords: ((“Prostate Cancer”[Mesh]) AND “multiparametric MRI”[Mesh]) AND “biparametric MRI”[Mesh]. As there are so many published papers on mpMRI, we decided to focus on the most relevant reviews and meta-analyses, which cover the majority of the studies.

The PRISMA diagram below shows the following steps in the inclusion process of the articles (Figure 2). The diagram was made using the draw.io application (United Kingdom).

All articles were included in a Microsoft Excel table for better management of this study, with information for the following columns: title, authors, used imaging method (mpMRI, bpMRI, or both), year of publication and journal, type of publication, and keywords.

We discussed the most important data and results, focusing on the pros and cons of each imaging technique and their limitations, and we grouped the articles as follows:The importance of mpMRI in PCa management—8 selected articles; some of the most relevant, of date articles regarding the role and limitations of mpMRI were selected and compared with one another.Comparison between mpMRI and bpMRI regarding PCa management—23 selected articles; we choose the most relevant studies, that compare the two imaging techniques regarding diagnosis, treatment, and follow-up of patients with PCa.

A table of the abbreviations used in the manuscript is listed below in alphabetical order to aid comprehension (Table 1).

## 3. Results

### 3.1. The Importance of mpMRI in PCa Management

Kumar et al. sustain that mpMRI plays an important role in evaluating PCa, by utilizing the different tissue properties of the prostate through its sequences [30]. PCa can be detected, localized, and characterized using these parameters, which provides both anatomical and functional information about the organ. The authors also suggest that the use of this imaging tool can reduce unnecessary biopsies, and also avoid false negative results, which is supported by studies conducted by Stabile et al. and Boesen et al. [20,31]. The authors conclude in their reviews that mpMRI has a high diagnostic accuracy for PCa. The American Urologic Association (AUA) recommends using mpMRI ahead of biopsy because it has higher sensitivity and specificity than PSA in screening for PCa [32]. Prior to mpMRI, PSA was used to determine whether a patient required a biopsy, and some of them were unnecessarily exposed to biopsy risks like hematuria, sepsis, and urinary retention [33].

This imaging technique can also be used for targeted biopsy; in this case, mpMRI can be used in three ways—cognitive fusion, ultrasound (US)-MRI fusion, and also MRI-MRI fusion—also known as in-core biopsy [30,34]. Compared to random biopsy, mpMRI-targeted biopsies improve PCa detection rates (from 21% to 43% in some studies) [20,30,32]. According to the PROMIS study, mpMRI has a higher sensitivity than TRUS-guided biopsy (93% vs. 48%), as well as a higher NPV (89%vs. 74%) [35]. Another advantage of using mpMRI biopsy is that it reduces the number of biopsy cores required [30]. For patients who require another evaluation after biopsy for any reason, it is recommended that the next mpMRI will be performed at least six weeks after the biopsy, because inflammation and bleeding can cause artifacts, resulting in false positive results [34].

Regarding active surveillance (AS), Kumar et al. state that mpMRI is not included in the treatment guidelines, but that prostate evaluation with mpMRI prior to biopsy will increase the number of patients who can remain on AS. Some studies indicate that it is effective in detecting PCa and rarely misses high-grade lesions, so it can be used as a tool in AS follow-ups [20,30,32]. Researchers suggest the fact that using mpMRI in follow-ups will reduce the number of annual biopsies in patients if the suspicious lesion does not increase of PIRADS [34].

Another application of mpMRI is in focal therapy, where it has a high sensitivity in localizing the suspect lesion [30]. Another important role to mention is in the management of radical prostatectomy [32]. When vascularization or new contrast enhancement is seen in the area anteriorly operated, mpMRI suggests recurrence [34].

One of the most significant limitations of mpMRI is that some clinically significant prostatic lesions can be missed by MRI which is why there is still concern about using mpMRI as a screening tool for PCa [32]. Other research indicates that mpMRI can easily miss apical lesions, especially when they are small in volume [34]. In terms of patients, absolute/relative contraindications to performing mpMRI include incompatible implants, severe claustrophobia, and previous severe reactions to gadolinium contrast medium [31].

The most important advantages and disadvantages of mpMRI can be found in the following table (Table 2), as long as its utility in different stages of PCa.

### 3.2. The Comparison between mpMRI and bpMRI Regarding PCa Management

In a meta-analysis conducted by Niu et al., it was found that bpMRI has a high sensitivity—81% and also a high specificity, of 77% in detecting PCa, but it does not use DCE images, which can identify subtle lesions that are easily missed when using non-contrast sequences [37]. Choi et al. conclude that bpMRI is appropriate for detecting clinically significant PCa, whereas DCE (mpMRI) assists in increasing PI-RADS from 3 to 4 in the PZ [38]. Furthermore, bpMRI reduces acquisition time and costs, as well as adverse reactions caused by contrast medium [38,39,40,41]. Pecoraro et al. also suggest that bpMRI can be a reliable tool in PCa diagnosis, with the exception of PI-RADS 3 and 4, where a DCE sequence is required for a more accurate evaluation of the lesion and a better classification [39,42,43]. As a disadvantage of using bpMRI, Pecoraro, and collaborators mention the inability of this technique to local stage PCa [39]. Many authors, however, maintain that the DCE sequence can be omitted because its role is minor, only for equivocal lesions, and that bpMRI and mpMRI have comparable accuracies in detecting PCa [41,44,45].

Boesen et al. conclude in their prospective study that bpMRI has a high NPV and can be used for PCa screening [46]. In the aforementioned study, 305 (30%) of 1020 men avoided biopsies using bpMRI, having low-risk lesions that required surveillance [46]. Through their research, Thestrup et al. confirm that bpMRI could be useful for AS [47]. The authors also state that the acquisition time for bpMRI is of approximately 15 min, while for mpMRI is of 30 min, plus 15 min for the intravenous administration of the contrast agents. This results in two bpMRI/hour and only one mpMRI/hour [47]. In terms of fusion biopsy, Sherrer et al. note that bpMRI can also be used to guide MRI/US fusion-targeted biopsy, comparable results to mpMRI [48].

Woo and collaborators conclude in a meta-analysis that, when compared to mpMRI, bpMRI has similar sensitivity (74%—bpMRI, 76%—mpMRI) and specificity (90%—bpMRI, 89%—mpMRI) in the diagnosis of PCa [49]. Alabousi et al. and Campli et al. found similar results in their studies [50,51]. Although most of the studies comparing bpMRI, and mpMRI used the same sequences, Cho et al. studied the comparison between these two imaging techniques using two different MRIs, taken at different times, with similar results—bpMRI can replace mpMRI in the detection of PCa, but its role in staging and extraprostatic extension detection is limited [52,53]. In terms of PPV in screening, Wallstorm et al. report that outperforms mpMRI (65% vs 62%) [54].

Caglic et al. compared the specificity and sensitivity of mpMRI to that of bpMRI in detecting extracapsular extension and seminal vesicle invasion. Contrary to the findings on PCa diagnosis, the results, show that bpMRI is not as accurate as mpMRI in these cases [55]. The results are shown in the table below (Table 3).

Other authors, however, such as Franco et al., maintain that bpMRI has a limited role in PCa management when compared to mpMRI [56]. Some of the arguments include lower image quality in bpMRI and as previously mentioned, the inability to distinguish between PI-RADS 3 or 4 [56].

The table below shows the sensitivity and specificity of bpMRI and mpMRI, respectively in detecting PCa (Table 4).

Based on the table above, the overall mpMRI sensitivity and specificity are 83% and 81%, while the overall sensitivity and specificity for bpMRI are 82% and 81%, respectively. We can observe that comparing mpMRI and bpMRI, the data are similar, with very little difference regarding sensitivity.

## 4. Discussion

Although tissular biopsy remains the “gold standard” for PCa diagnosis, MRI has been used to precisely identify and evaluate suspicious lesions in recent years [58]. Imaging techniques improved the grading of suspicious prostate lesions allowing clinicians to recommend treatment based on cancer prognosis and making watchful waiting or active surveillance a therapeutic option for patients with low-risk lesions. Although transrectal prostate ultrasound (TRUS) is a valuable imaging tool for assessing prostate size and anatomy and can be used to diagnose PCa, it does not provide enough details for grading and staging, and the reports with the adjacent tissues are not very accurate [59,60].

MpMRI has been used in the management of PCa for many years, from screening and diagnosis to surveillance, biopsy, and treatment [61,62,63]. The cost-effectiveness of this method, however, has become a worldwide concern, and specialists are beginning to look for more feasible methods that could overcome this imaging technique at a lower cost. Another issue raised by many researchers the acute adverse reactions caused by gadolinium contrast agents. The most serious of these are nausea, vomiting, dizziness, headache, seizure, pulmonary edema, and anaphylaxis [64], reactions that, while rare, can put the patients in real danger, given that the majority of patients have associated pathological conditions due to PCa and its treatment, other comorbidities, and age. The presence of end-stage renal disease, as well as acute kidney injury, chronic kidney disease, and dialysis, is another issue related to contrast agents [65,66]. The majority of the concerns raised in studies regarding gadolinium-based contrast agents (GBCA) are related to nephrogenic systemic fibrosis [66]. This pathology manifests as a progressive multiorgan fibrosing condition caused by GBCA, and it primarily involves the thickening of the organism’s fibrous tissue, including the liver, heart, lungs, and muscles [67]. As a general rule, GBCA imaging studies should be avoided in patients with an estimated glomerular filtration rate of less than 30 mL/min/1.73 m2, unless no non-contrast MRI imaging techniques are available [68]. In the response to the aforementioned issues, specialists have proposed using bpMRI instead of mpMRI, because it does not require contrast agent administration, which is advantageous in terms of cost-effectiveness, exclusion of adverse reactions, and also time-effectiveness [42,57]. As bpMRI does not require the acquisition of contrast agents, it is less expensive than mpMRI. In countries such as the United States of America, the costs vary depending on the healthcare services provided, whereas in Korea, for example, bpMRI costs half as much as mpMRI [65].These recommendations and suggestions emphasize that the importance of using bpMRI instead of mpMRI whenever possible.

Other contraindications to performing an MRI on patients include cardiac implantable electronic devices (such as pacemakers and defibrillators), metallic intraocular foreign bodies, cochlear implants, artificial limbs, and other materials incompatible with the MRI [69]. In addition to the conditions mentioned above, claustrophobia and extremely high weights can make the MRI scan problematic [70]. As they have nothing to do with the administration of contrast agents, all of these issues are also available when using bpMRI instead of mpMRI.

Regarding AS, recent research suggests that biopsies are not yet safe to be abandoned in the absence of MRI progression [71]. Patients who are eligible for AS and those who are arleady on AS will undergo mpMRI scans, which will reduce the need for repeated biopsies [72,73]. There are currently more studies attempting to define the role of bpMRI in AS [74].

Most research papers on this subject reach the same conclusion: bpMRI could replace mpMRI in the diagnosis of clinically significant PCa, with nearly the same accuracy in the detection of suspicious lesions, with one exception—lesions graded PI-RADS 3 and 4, which require DCE sequence as well. The DCE sequence is required because a PI-RADS 3 lesion indicates an equivocal lesion with a significant chance of developing clinically significant prostate cancer, whereas a PI-RADS 4 lesion indicates that clinically significant cancer is likely to be present [26]. Researchers state that using DCE for PI-RADS 3 can change the grading to PI-RADS 4 if the enhancement is focal [75]. Another role for bpMRI that has been mentioned is for AS, as long as the lesion does not upgrade in terms of PI-RADS. A DCE sequence is also required for extracapsular extension, as Caglic et al. pointed out in their study [54]. As a result, mpMRI remains the preferred imaging modality. In terms ofPCa diagnosis, our overall results in comparing mpMRI and bpMRI sensitivity and specificity are comparable to the ones found in the literature, with no significant difference between these two methods.

Concerning fusion biopsies, and treatment, as long as evaluating the operated site for recurrence after prostatectomy, there are no sufficient studies and data to make an accurate comparison between these two imaging methods. Only a few studies have concluded that bpMRI can be used instead of mpMRI to guide MRI/US fusion-targeted biopsy [48]. Some authors argue that, while mpMRI-ultrasound fusion biopsy is associated with high cancer detection rates in clinically significant PCa, it has some limitations that TRUS biopsies alone do not have [74,76].

Some authors state that when scanning for recurrence, mpMRI cannot be surpassed by bpMRI because the DCE sequence is required. According to the arguments, recurrence is suspected when new vessels appear in the previously operated area, a situation that can be better evaluated when a contrast medium is used.

Despite all of the advantages mentioned above regarding PCa detection, mpMRI is currently the imaging method of choice because bpMRI is still in trials and does not yet have a concrete protocol that can be implemented globally. MpMRI, on the other hand has a high accuracy for detecting local recurrence, even in patients with low PSA Levels [77]. Another important fact that requires further investigation is the value of DCE on different zones of the prostate, in order to better understand whether bpMRI is more appropriate in different zones of the organ.

Above and beyond the advantages and disadvantages of cost-effectiveness, the most important aspect to consider when selecting an imaging method is the benefits that it provides to the patient.

## 5. Conclusions

BpMRI is a valuable imaging tool in the management of PCa, with similar accuracies in diagnosis as mpMRI. Furthermore, it reduces the high costs of mpMRI as well as the adverse reactions caused by contrast agents. Nonetheless, its role in detecting extracapsular invasion, AS, and follow-up is still inferior to mpMRI. More research is needed to determine whether bpMRI can replace mpMRI in the steps of PCa management.

### 5.1. Limitations of the Study

Unfortunately, of the majority of papers on bpMRI examine its accuracy only in the the diagnosis of PCa. We did not have enough data to compare bpMRI and mpMRI in terms of AS and treatment.

### 5.2. Future Directions

Some future directions include determining the role of bpMRI in AS, biopsy guiding, local staging, and PCa treatment, as well as developing protocols for using bpMRI whenever possible, in the PCa management steps where it has comparable accuracy as mpMRI.

In order for the research to be objective, a large number of patients who had mpMRI should have bpMRIs read by another experienced radiologists. The results should be compared in order to better understand what the limitations of bpMRI are and which steps in PCa management should be taken. As previously stated, the different zones of the prostate should be studied separately in order to obtain more accurate results on the accuracy and roles of bpMRI, as well as its limitations.

## Figures and Tables

**Figure 1 life-13-00465-f001:**
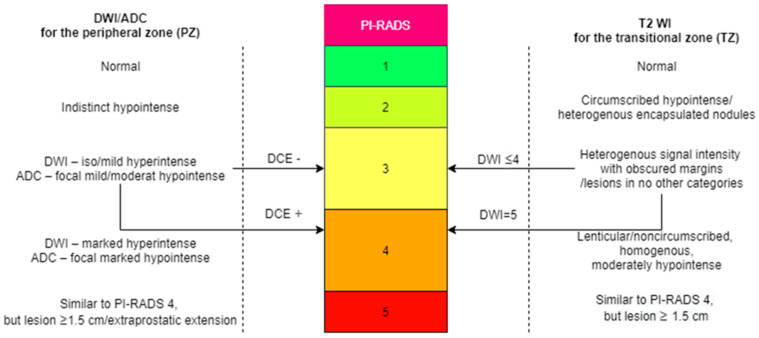
PI−RADS v2 for mpMRI [17]. The colors were used to distinguish the normal prostate, which has a very low risk of cancer (green), from benign lesions, which have a low and intermediate risk of cancer (the two types of yellow), from those that have a high risk of being malignant (orange), and a very high risk of being malignant (red).

**Figure 2 life-13-00465-f002:**
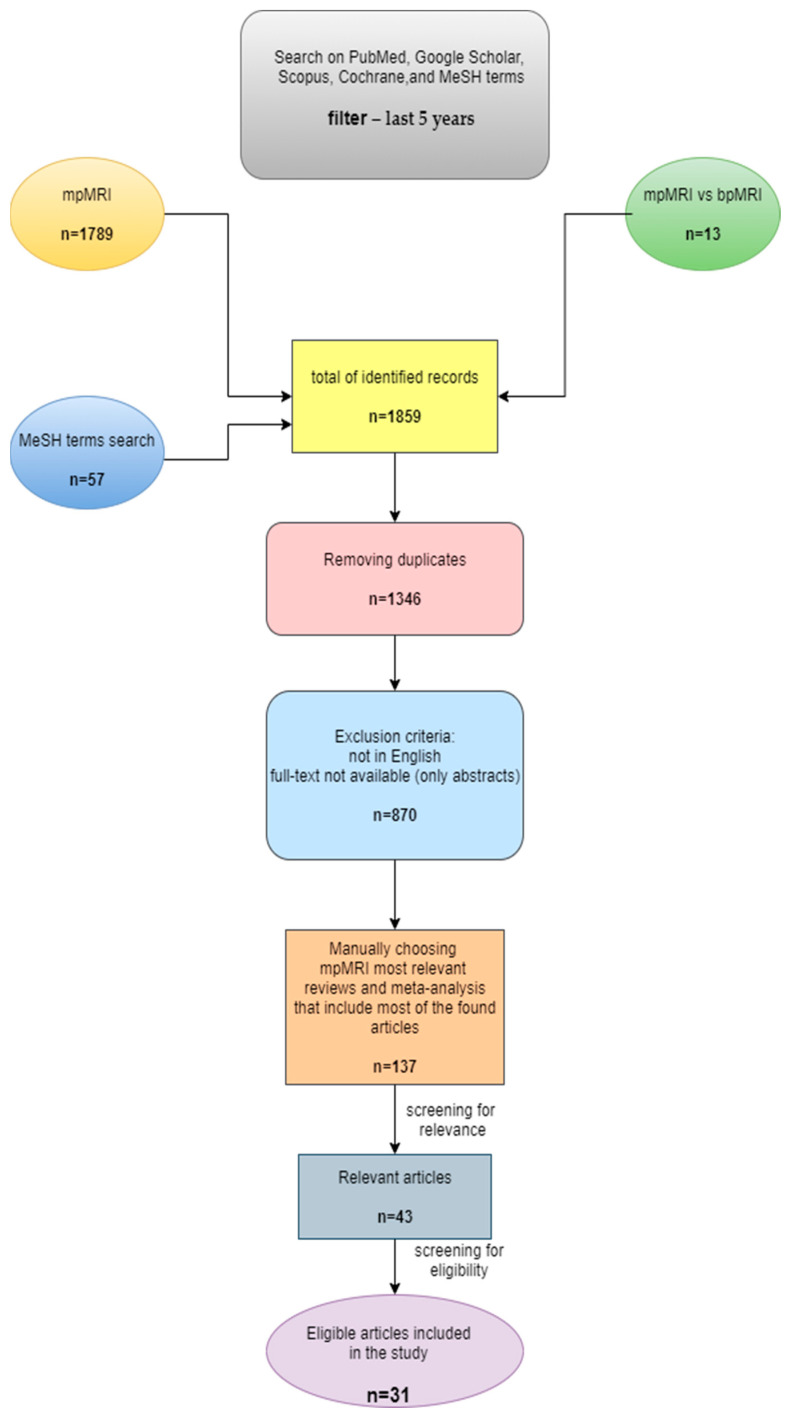
PRISMA chart of the article selection.

**Table 1 life-13-00465-t001:** List of abbreviations.

Abbreviation	Terms
AS	Active surveillance
BPH	Benign prostate hyperplasia
bpMRI	Biparametric MRI
CZ	Central zone
DWI	Diffusion-weighted images
DCE/DCEI	Dynamic contrast-enhanced imaging
ESUR	European Society of Urogenital Radiology
GBCA	Gadolinium-based contrast agents
MRI	Magnetic resonance imaging
mpMRI	Multiparametric MRI
NPV	Negative predictive value
PZ	Peripheral zone
PET-CT	Positon emission tomography and computed tomography
PPV	Positive predictive value
PRISMA	Preferred Reporting Items for Systematic Reviews and Meta-Analysis
PCa	Prostate cancer
PI-RADS	Prostate Imaging-Reporting and Data System
PSA	Prostate-specific antigen
TZ	Transition zone
TRUS	Transrectal ultrasound
US	ultrasound

**Table 2 life-13-00465-t002:** Advantages and disadvantages of mpMRI.

No.	Title	Authors	Year of Public.	Advantages of Method	Disadvantages of Method
1	Multiparametric (mp) MRI of Prostate Cancer	V. Kumar et al. [30]	2018	high negative predictive value; detection and localization of suspicious lesions—decrease sample error of biopsy; reduce unnecessary biopsies;	adverse reactions to contrast agents
2	Multiparametric MRI for prostate cancer diagnosis: current status and future directions	A. Stabile et al. [20]	2020		less frequently used in non-academic centers; disagreements, even among experienced radiologists.
3	Use of multiparametric magnetic resonance imaging (mpMRI) in active surveillance for low-risk prostate cancer: a scoping review on the benefits and harm of mpMRI in different biopsy scenarios	K. Chiam et al. [36]	2021		more limited in active surveillance than in detection of Pca
4	Use of multiparametric magnetic resonance imaging (mpMRI) in localized prostate cancer	L. O’Connor et al. [32]	2020	more accurate visualization and sampling of localized prostate cancer; advancements in focal therapy for prostate cancer;	utility for excluding prostate cancer is still unclear; delay in the treatment of clinically significant MRI “invisible” lesions
5	Role of MRI in the detection of prostate cancer	R.C. Wu et al. [33]	2021	increasing the sensitivity to detect clinically significant prostate cancer; limiting the diagnosis of small-volume, low-risk cancers	
6	Multiparametric magnetic resonance imaging: Overview of the technique, clinical applications in prostate biopsy, and future directions	H. C. Demirel et al. [34]	2018	the use of surveillance reduces the number of annual biopsies as long as the lesion does not increase in PIRADS; local recurrence after radical prostatectomy—mpMRI may show vascularization and contrast enhancement in the operation area	64% of multifocal cancers were found in the final pathology of RP materials, whereas mpMRI was only able to detect 21% of these foci
7	Multiparametric MRI to improve detection of prostate cancer compared with transrectal ultrasound-guided prostate biopsy alone: the PROMIS study	L.C. Brown et al. [35]	2018	less overdiagnosis; less overtreatment	studies mostly evaluate only the peripheral zone of the prostate
8	Multiparametric MRI in the detection and staging of prostate cancer	L. Boesen [31]	2018	can improve the detection rate of clinically significant Pca	contraindications: non-MRI compatible pacemakers, magnetic implants, severe claustrophobia, previous moderate or severe reactions to gadolinium-based contrast substances

**Table 3 life-13-00465-t003:** The specificity and sensitivity of mpMRI vs. bpMRI regarding extracapsular extension and seminal vesicle invasion.

	mpMRI Sensitivity	mpMRI Specificity	bpMRI Sensitivity	bpMRI Specificity
extracapsular extension	66%	84%	59%	87%
seminal vesicle invasion	83%	97%	66%	92%

**Table 4 life-13-00465-t004:** Comparison between the sensitivity and specificity of bpMRI and mpMRI in detecting PCa.

No.	Article Title	Authors	mpMRI Sensitivity	mpMRI Specificity	bpMRI Sensitivity	bmMRI Specificity
1	Diagnostic Performance of Biparametric MRI for Detection of Prostate Cancer: A Systematic Review and Meta-Analysis	Niu et al. [37]	85%	77%	80%	80%
2	Comparison of bi- and multiparametric magnetic resonance imaging to select men for active surveillance	Thesthrup et al. [47]	76%	89%	74%	90%
3	Biparametric versus Multiparametric Prostate MRI for the Detection of Prostate Cancer in Treatment-Naive Patients: A Diagnostic Test Accuracy Systematic Review and Meta-Analysis	Alabousi et al. [50]	85%	74%	88%	72%
4	Evolution of prostate MRI: from multiparametric standard to less-is-better and different-is better strategies	Girometti et al. [53]	94%	84%	93%	87%
5	Comparison of biparametric and multiparametric MRI in the diagnosis of prostate cancer	Xu et al. [57]	77%	84%	76%	79%

## Data Availability

No new data were created or analyzed in this study. Data sharing is not applicable to this article.

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
