# Peer review of "Could Biparametric MRI Replace Multiparametric MRI in the Management of Prostate Cancer?"

_life, 2023, doi:10.3390/life13020465_

Round 1

Reviewer 1 Report

life-2192730

The authors present a review that compairs the mpMRI to the bpMRI and discusses the ability of performing bpMRI instead of mpMRI for the initial diagnosis for PCa. Although I am not english speaking myself, the manuscript and the abstract lack in english expression. Careful editing with the assistance of a native english speaking person should be considered by the authors.

 The structure is correct but some questions were raised while studying it.

Simple Summary

17. “Tumors” instead of “tumor”. 

19. “Until it advances or it is advanced” instead of “its advanced”. Also “vital” instead of “vitally”.

24. “disadvantages” is correct.

Abstract

31.  “Small cancer” should be rephrased. Small according to size or grade?

34. “and in treatment”. “in” could be removed.

38-39. “it’s role” sounds better.  “A single paragraph of about 200 words 39

maximum.” Please remove this sentence.

Introduction

46. “tumors” instead of “tumor”.  

47. “deaths” instead of “death”.

52. Subperitoneally” instead of “subperitoneal”.

55. “supportive role in the male reproductive system.” Please fill the phrase.

71. “transrectally” instead of “transrectal” and “trans-perineally” instead of “trans-perineal”.

72. “positron” instead of “positon”.

114-117. “Using PubMed, Cochrane Library, Google Scholar, and Scopus, we selected the most important studies in the field, that compare mpMRI with bpMRI, but also evaluate bpMRI alone. After using different filters and after a manual selection of the papers, we selected 31 articles eligible for the study.” This paragraph belongs to “Material and Methods”.

Materials and methods

119: “PRISMA” should be referred as citation.

Discussion

167. “in core” instead of “in bore”.

172. “Reason” instead of “reason”.

178-179. “Some studies suggestcy in detecting PCa, and rarely misses high-grade lesions”. Please rephrase and explain.

188-190. “One of the most important limitations of mpMRI is that some prostatic lesions that are clinically significant, can be missed by MRI, reasons why there remains a concern about using mpMRI as a screening tool for PCa”. Please rephrase.

200. “Meta-analysis” instead of “meta-analyze”.

Sensitivity parameters etc are either described as % or as e.g. 0.81. It would be better if you described them in one manner.

Also, a table with abbreviations should be presented.

Finally, it is a review that needs work to improve. Also, it describes a path that could be used in specific patients and help reducing the cost.

Reviewer 2 Report

This review paper summarized the pros and cons of using biparametric MRI (bpMRI) in replacement of multiparametric MRI (mpMRI) in prostate cancer (PCa) management from related literatures. The conclusion is that while bpMRI reduces the cost and time and adverse reactions caused by the contrast agents that are necessary in mpMRI, the accuracy is only demonstrated comparable in PCa diagnosis and not in other aspects of disease management. The reviewer suggests the following comments and questions be addressed before the article is suitable for publication:

1.     A few typos:

a.     Line 19 “…it is vitally to diagnose…”

b.     Line 39, “a single paragraph…”

c.     Line 172, “for some reaso, …”

2.     In line 104 and a few other places in the article, the authors mentioned thee bpMRI is faster is acquisition. Is there any quantitative comparison of how much time is saved? This will help the readers to better assess whether the value of the replacement.

3.     In line 259, the reviewer suggests to also mention the efficiency in term of time in addition to the cost-effectiveness of bpMRI.

4.     Needs clarification in line 297: are PI-RADS 3 and 4 malignant or benign?

5.     The review article lacks a paragraph of elaboration of what needs to be done to further assess bpMRI to replace mpMRI. For example, what aspects of assessments needs to be focused on to draw a more solid conlusion, and what are the guidelines for bpMRI development to make it more suitable in the PCa management, etc.
